# A Domain-Shrinking based Bayesian Optimization Algorithm with Order-Optimal Regret Performance

**Sudeep Salgia**[*], **Sattar Vakili**[†], **Qing Zhao**[*]
[*]School of Electrical and Computer Engineering
Cornell University, Ithaca, NY
{ss3827,qz16}@cornell.edu
[†]MediaTek Research, UK
sattar.vakili@mtkresearch.com

## Abstract

We consider sequential optimization of an unknown function in a reproducing kernel Hilbert space. We propose a Gaussian process-based algorithm and establish its order-optimal regret performance (up to a poly-logarithmic factor). This is the first GP-based algorithm with an order-optimal regret guarantee. The proposed algorithm is rooted in the methodology of domain shrinking realized through a sequence of tree-based region pruning and refining to concentrate queries in increasingly smaller high-performing regions of the function domain. The search for high-performing regions is localized and guided by an iterative estimation of the optimal function value to ensure both learning efficiency and computational efficiency. Compared with the prevailing GP-UCB family of algorithms, the proposed algorithm reduces computational complexity by a factor of $O(T^{2d-1})$ (where $T$ is the time horizon and $d$ the dimension of the function domain).

## 1   Introduction

Consider a black-box optimization problem with an unknown objective function $f : \mathcal{X} \to \mathbb{R}$, where $\mathcal{X} \subset \mathbb{R}^d$ is a convex and compact set. The learner can access the function only through a noisy oracle, which, when queried with a point $x \in \mathcal{X}$, returns a noisy function value at that point. The learning objective is to approach the maximizer $x^*$ of the function through a sequence of query points $\{x_t\}_{t=1}^T$ chosen sequentially in time. The learning efficiency is measured by cumulative regret given by

$$R(T) = \sum_{t=1}^{T} \left[ f(x^*) - f(x_t) \right]. \tag{1}$$

This cumulative regret measure dictates the online nature of the problem: every query point during the learning process carries loss, not just the end point $x_T$ after learning concludes. The classical exploration-exploitation tradeoff in online learning hence ensues.

### 1.1   Gaussian Process Models

The above problem is ill-posed unless certain structure of the unknown objective function $f$ is assumed to make learning $x^*$ feasible. One such structural assumption is the convexity of $f$, which leads to the class of stochastic convex optimization problems. Another class of black-box optimization problems that is gaining interest in recent years is kernel-based learning where $f$ is assumed to live in a Reproducing Kernel Hilbert Space (RKHS) associated with a positive-definite kernel. An effective approach to kernel-based black-box optimization is Bayesian optimization that adopts a *fictitious* prior on the unknown function $f$. In other words, while $f$ is deterministic, it is viewed internally by

35th Conference on Neural Information Processing Systems (NeurIPS 2021).

the learning algorithm as a realization of a random process over $\mathcal{X}$. A natural choice is the Gaussian process (GP) with a Gaussian prior due to the conjugate property that significantly simplifies the analytical form of the posterior distribution at each newly obtained observation.

In a celebrated work, Srinivas *et al.* [1] proposed the GP-UCB algorithm that constructs a proxy of $f$ using the upper confidence bound (UCB) concept first introduced in the classical multi-armed bandit problem [2, 3]. Specifically, at each time instant $t$, a UCB of $f$ is constructed using the closed-form posterior mean and standard deviation of the GP model of $f$. The algorithm then sets the next query point to be the maximizer of the UCB. Several variations of GP-UCB, tailored for different settings (see Sec 1.3), have since been developed.

The GP-UCB family of algorithms generally enjoy good empirical performance in terms of regret. The analytical guarantees of their regret performance, however, leave considerable gaps to the existing lower bound [4]. More significantly, the state-of-the-art regret bound of GP-UCB does not guarantee a sublinear order in $T$ for certain kernels, hence a lack of guaranteed convergence to $f(x^*)$ [4, 5].

Another difficulty with the GP-UCB family of algorithms is their computational complexity, which can be prohibitive as the dimension $d$ and/or the horizon length $T$ grows. The computational complexity has two main sources: (i) the inversion of the covariance matrix in updating the posterior GP distribution, which has an $O(t^3)$ complexity with $t$ samples; (ii) the maximization of the UCB proxy over the entire domain $\mathcal{X}$ at each time instant. In particular, due to the multi-modality of the UCB score, its maximization is often carried out using a grid search with an increasingly finer discretization of the entire domain. Specifically, due to analytical requirements, the discretization is typically assumed to grow in the order of $O(t^{2d})$ [1, 6], resulting in an overall computational complexity of $O(T^{2d+3})$.

Several studies exist that tackle the first source of high complexity of GP-UCB, using sparse matrix approximation techniques to reduce the complexity in the inversion of the covariance matrix (see, e.g., [7, 8]). The second source, which is the dominating factor, has not been effectively addressed.

## 1.2 Main results

The goal of this work is to develop a GP-based Bayesian optimization algorithm with a regret guarantee that closes the gap to the lower bound. Furthermore, we tackle the second source of the complexity to ensure both learning efficiency and computational efficiency.

Referred to as GP-ThreDS (Thresholded Domain Shrinking), the proposed algorithm is rooted in the methodology of domain shrinking: it continuously prunes sub-performing regions of the domain $\mathcal{X}$ and zooms into increasingly smaller high-performing regions of $\mathcal{X}$ as time goes. The purpose of the domain shrinking is twofold. First, it ensures high learning efficiency by focusing queries on regions of $\mathcal{X}$ with function values approaching $f(x^*)$. Second, it achieves computational efficiency by avoiding a global maximization of the proxy function over the entire domain $\mathcal{X}$.

Our specific approach to domain shrinking is built upon a sequence of localized searches on a growing binary tree that forms successively refined partitions of $\mathcal{X}$. Starting from the root of the tree that represents the entire domain, the search progresses down the tree by adaptively pruning nodes that do not contain the maximizer with high probability, consequently zooming into increasingly smaller high-performing regions of $\mathcal{X}$ as the search deepens. Another progressive thread in this sequence of localized searches is the criterion for pruning the tree. Each localized search aims to identify nodes at a certain depth of the tree that contain points with function values exceeding a given threshold. The threshold is updated iteratively to approach the maximum function value $f(x^*)$. More succinctly, the proposed algorithm is a sequence of localized searches in the domain of the function guided by an iterative search in the range of the function.

The above domain shrinking approach via localized search is the primary contributing factor to improved performance in terms of both regret guarantee and computational complexity. In particular, the rate of domain shrinking is controlled to ensure not only the concentration of query points in high-performing regions, but also a *constant*-sized discretization at all times when estimating the function values. This constant-sized discretization allows a tighter regret analysis and results in a regret upper bound for GP-ThreDS that matches with the lower bound (up to a poly-logarithmic factor). We show that the regret of GP-ThreDS is $O(\sqrt{T\gamma_T})$ (up to a poly-logarithmic factor), where $\gamma_T$ denotes the maximum information gain after $T$ steps and is representative of the *effective* dimension of the

problem [9, 10]. In the case of Matérn and Squared Exponential (SE) kernels where the lower bounds on regret are known, on substituting the improved bounds on $\gamma_T$ from [11], our results match the lower bounds and close the gap reported in [4, 12]. In comparison, the state-of-the-art analysis of GP-UCB yields an $O(\gamma_T\sqrt{T})$ regret bound [e.g., see, 6, Theorem 3]. The $O(\sqrt{\gamma_T})$ gap between the regret guarantees of GP-UCB and the proposed GP-ThreDS is significant: it can grow polynomially in $T$ (e.g. in the case of Matérn kernel).

Computation-wise, the constant-sized discretization contrasts sharply with the growing (at rate $O(t^{2d})$ with time $t$) discretization required by the GP-UCB family of algorithms. Another factor contributing to the reduced complexity is the relaxed search criterion that aims to determine only the existence of threshold-exceeding points, in contrast to finding a global maximizer as in the GP-UCB family of algorithms. As a result, GP-ThreDS reduces the computational complexity from $O(T^{2d+3})$ as required by GP-UCB family of algorithms to $O(T^4)$.

## 1.3 Related Work

There is a vast body of literature on numerical and theoretical analysis of Bayesian optimization algorithms. With our focus on a computationally efficient algorithm with a provable regret guarantee, the most relevant results to ours are [1] and [6] discussed above. [6] also proved the same $O(\gamma_T\sqrt{T})$ regret holds for GP-TS, a Bayesian optimization algorithm based on Thompson sampling principle. Augmenting GP models with local polynomial estimators, [13] introduced LP-GP-UCB and established improved regret bounds for it under special cases [see, 13, Sec. 3.2]. However, for other cases, the regret guarantees for LP-GP-UCB remain in the same order as GP-UCB. More recently, [5] introduced $\pi$-GP-UCB, specific to Matérn family of kernels, that constructs a cover for the search space, as many hypercubes, and fits an independent GP to each cover element. This algorithm was proven to achieve sublinear regret across all parameters of the Matérn family. Almost all other algorithms in the GP-UCB family have a regret guarantee of $O(\gamma_T\sqrt{T})$, which is $O(\sqrt{\gamma_T})$ greater than the lower bound and can grow polynomially in $T$. Two exceptions to this are the SupKernelUCB and the RIPS algorithms proposed in [10] and [14] which achieve a regret of $O(\sqrt{T\gamma_T})$ for discrete action spaces. While this may be extendable to continuous spaces via a discretization argument as recently pointed out in [5, 12], the required discretization needs to grow polynomially in $T$, making it computationally expensive. Moreover, it has been noted that SupKernelUCB performs poorly in practice [5, 8, 12]. GP-ThreDS, on the other hand, is a computationally efficient algorithm that achieves tight regret bounds with good empirical performance (see Sec. 5). A comparison with other related works including the ones in different settings such as noise-free observations and random $f$ are deferred to the supplementary.

## 2 Problem Statement

### 2.1 Problem Formulation

We consider the problem of optimizing a fixed and unknown function $f : \mathcal{X} \to \mathbb{R}$, where $\mathcal{X} \subset \mathbb{R}^d$ is a convex and compact domain. A sequential optimization algorithm chooses a point $x_t \in \mathcal{X}$ at each time instant $t = 1, 2, \ldots$, and observes $y_t = f(x_t) + \epsilon_t$, where the noise sequence $\{\epsilon_t\}_{t=1}^{\infty}$ is assumed to be i.i.d. over $t$ and $R$-sub-Gaussian for a fixed constant $R \geq 0$, i.e., $\mathbb{E}\left[e^{\zeta\epsilon_t}\right] \leq \exp\left(\zeta^2 R^2/2\right)$ for all $\zeta \in \mathbb{R}$ and $t \in \mathbb{N}$.

We assume a regularity condition on the objective function $f$ that is commonly adopted under kernelized learning models. Specifically, we assume that $f$ lives in a Reproducing Kernel Hilbert Space (RKHS)[1] associated with a positive definite kernel $k : \mathcal{X} \times \mathcal{X} \to \mathbb{R}$. The RKHS norm of $f$ is assumed to be bounded by a known constant $B$, that is, $\|f\|_k \leq B$. We further assume that $f$ is $\alpha$-Hölder continuous, that is, $|f(x) - f(x')| \leq L\|x - x'\|^\alpha$ for all $x, x' \in \mathcal{X}$ for some $\alpha \in (0, 1]$ and $L > 0$. This is a mild assumption as this is a direct consequence of RKHS assumption for commonly used kernels as shown in [13]. We also assume the knowledge of an interval $[a, b]$, such

---

[1]The RKHS, denoted by $H_k$, is a Hilbert space associated with a positive definite kernel $k(\cdot, \cdot)$ and is fully specified by the kernel and vice versa. It is endowed with an inner product $\langle \cdot \rangle_k$ that obeys the reproducing property, i.e., $g(x) = \langle g, k(x, \cdot)\rangle_k$ for all $g \in H_k$. The inner product also induces a norm $\|g\|_k = \langle g, g\rangle_k$. This norm is a measure of the smoothness of the function $f$ with respect to the kernel $k$ and is finite if and only if $f \in H_k$.

that $f(x^*) \in [a, b]$. This is also a mild assumption as domain-specific knowledge often provides us with bounds. For example, a common application of black-box optimization is hyperparameter tuning in deep learning models. The unknown function represents the accuracy of the model for a given set of hyperparameters. Since $f$ represents the accuracy of the model, we have $f(x^*) \in [0, 1]$. For simplicity of notation, we assume $\mathcal{X} = [0, 1]^d$ and $f(x^*) \in [0, 1]$. It is straightforward to relax these assumptions to general compact domains and arbitrary bounded ranges $[a, b]$.

Our objective is a computationally efficient algorithm with a guarantee on regret performance as defined in (1). We provide high probability regret bounds that hold with probability at least $1 - \delta_0$ for any given $\delta_0 \in (0, 1)$, a stronger performance guarantee than bounds on expected regret.

## 2.2 Preliminaries on Gaussian processes

Under the GP model, the unknown function $f$ is treated hypothetically as a realization of a Gaussian process over $\mathcal{X}$. A Gaussian Process $\{F(x)\}_{x \in \mathcal{X}}$ is fully specified by its mean function $\mu(\cdot)$ and covariance function $k(\cdot, \cdot)$. All finite samples of the process are jointly Gaussian with mean $\mathbb{E}[F(x_i)] = \mu(x_i)$ and covariance $\mathbb{E}[(F(x_i) - \mu(x_i))(F(x_j) - \mu(x_j))] = k(x_i, x_j)$ for $1 \leq i, j \leq n$ and $n \in \mathbb{N}$ [15]. The noise $\epsilon_t$ is also viewed as Gaussian.

The conjugate property of Gaussian processes with Gaussian noise allows for a closed-form expression of the posterior distribution. Consider a set of observations $\mathcal{H}_t = \{\mathbf{x}_t, \mathbf{y}_t\}$ where $\mathbf{x}_t = (x_1, x_2, \ldots, x_t)^T$ and $\mathbf{y}_t = (y_1, y_2, \ldots, y_t)^T$. Here $y_s = f(x_s) + \epsilon_s$ where $x_s \in \mathcal{X}$ and $\epsilon_s$ are the zero-mean noise terms, i.i.d. over $s$ for $s \in \mathbb{N}$. Conditioned on the history of observations $\mathcal{H}_t$, the posterior for $f$ is also a Gaussian process with mean and covariance functions given as

$$\mu_t(x) = \mathbb{E}[F(x)|\mathcal{H}_t] = k_{\mathbf{x}_t,x}^T (K_{\mathbf{x}_t,\mathbf{x}_t} + \lambda I)^{-1} \mathbf{y}_t \tag{2}$$

$$k_t(x, x') = \mathbb{E}[(F(x) - \mu_t(x))(F(x') - \mu_t(x'))|\mathcal{H}_t] = k(x, x') - k_{\mathbf{x}_t,x}^T (K_{\mathbf{x}_t,\mathbf{x}_t} + \lambda I)^{-1} k_{\mathbf{x}_t,x'}. \tag{3}$$

In the above expressions, $k_{\mathbf{x}_t,x} = [k(x_1, x), \ldots, k(x_t, x)]^T$, $K_{\mathbf{x}_t,\mathbf{x}_t}$ is the $t \times t$ covariance matrix $[k(x_i, x_j)]_{i,j=1}^t$, $I$ is the $t \times t$ identity matrix and $\lambda$ is the variance of the Gaussian model assumed for the noise terms.

Gaussian processes are powerful non-parametric Bayesian models for functions in RKHSs [16]. In particular, the mean function of the GP regression (eqn. (2)) lies in the RKHS with kernel $k(\cdot, \cdot)$ with high probability. We emphasize that the GP model of $f$ and the Gaussian noise assumption are internal to the learning algorithm. The underlying objective function $f$ is an arbitrary deterministic function in an RKHS, and the noise obeys an arbitrary $R$-sub-Gaussian distribution.

# 3  The GP-ThreDS Algorithm

In Sec. 3.1, we present the basic domain-shrinking structure of GP-ThreDS that continuously prunes sub-performing regions of $\mathcal{X}$ and zooms into increasingly smaller high-performing regions of $\mathcal{X}$. In Sec. 3.2, we present the method for identifying high-performing regions of $\mathcal{X}$.

## 3.1  Thresholded domain shrinking

GP-ThreDS operates in epochs. Each epoch completes one cycle of pruning, refining, and threshold updating as detailed below. **(i) Pruning:** removing sub-performing regions of $\mathcal{X}$ from future consideration; **(ii) Refining:** splitting high-performing regions of $\mathcal{X}$ into smaller regions for refined search (i.e., zooming in) in future epochs; **(iii) Threshold updating:** updating the threshold on function values that defines the criterion for high/sub-performance to be used in the next epoch. The pruning and refining conform to a binary-tree representation of $\mathcal{X}$ with nodes representing regions of $\mathcal{X}$ and edges the subset relation (i.e., region splitting). Throughout the paper, we use nodes and regions of $\mathcal{X}$ interchangeably.

We explain the details with an example. Consider a one-dimensional function over $\mathcal{X} = [0, 1]$ as shown in Fig. 1. Assume that it is known $f(x^*) \in [0, 1.4]$. The function threshold $\tau_1$ defining the pruning criterion in the first epoch is set to the mid-point: $\tau_1 = 0.7$. In epoch 1, the domain $\mathcal{X}$ is represented by a tree of height 1 with the root representing the entire domain $[0, 1]$ and the two leaf

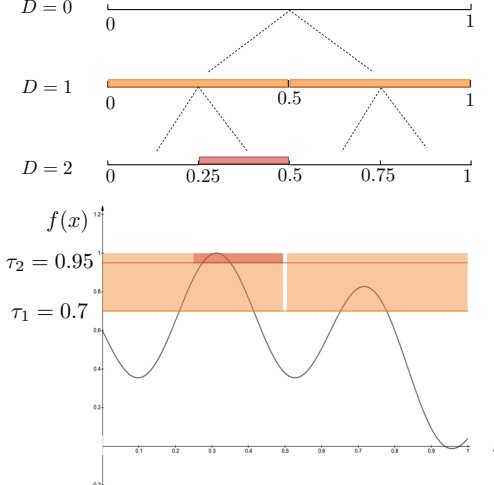

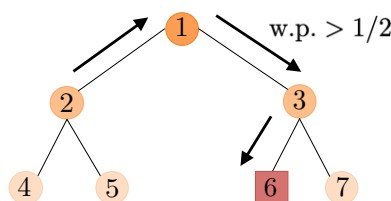

Figure 2: An illustration of the random-walk based search. (Node 6 is the single high-performing leaf node. If the random walk is currently at node 2, the correct direction is along the shortest path to node 6: via node 1 and then node 3.)

Figure 1: Thresholded domain shrinking.

nodes representing the two sub-intervals $[0, 0.5]$ and $(0.5, 1]$ (see Fig. 1). In the pruning stage of this epoch, the algorithm determines, with a required confidence, whether each leaf node contains a point with function value exceeding $\tau_1$. Such threshold-exceeding leaf nodes are referred to as high-performing nodes. Otherwise, they are called sub-performing nodes and are pruned, along with their ancestors, from the tree. Suppose that in this example, both sub-intervals $[0, 0.5]$ and $(0.5, 1]$ are identified as high-performing (see Sec. 3.2 on identifying high-performing nodes). Consequently, no node is pruned, and the algorithm proceeds to the refining stage, where each sub-interval splits, and the tree grows to a height of 2 with four leaf nodes. The threshold is then updated to $\tau_2 = 0.95$ (see below on threshold updating). The increased threshold reflects an adjustment toward a more aggressive pruning in the next epoch as suggested by the presence of (multiple) high-performing nodes in the current epoch.

In the second epoch, the pruning stage aims to identify high-performing (defined by $\tau_2$) nodes among the four leaf nodes. Supposed that it is determined leaf node $(0.25, 0.5]$ is the only high-performing node. Then the nodes $[0, 0.25], (0.5, 0.75]$ and $(0.75, 1]$ and all their ancestors are pruned. In the refining stage, the high-performing node $(0.25, 0.5]$ splits into two. The threshold is updated to $\tau_3$. The algorithm then progresses into the third epoch, facing the same decision problem on the two leaf nodes (the two children of $(0.25, 0.5]$) of the pruned tree and following the same pruning-refining-threshold updating cycle.

For a general $d$-dimensional problem, the basic structure is the same with three simple generalizations. First, the two children of any given node are formed by equally splitting the longest edge of the corresponding $d$-dimensional cuboid (ties broken arbitrarily). Second, in each epoch, the tree grows by $d$ levels ($d = 1$ in the above example) in the refining stage by following successive binary splitting $d$ times. The last detail to specify is that if no leaf node is identified as high-performing in an epoch $k$, then the refining stage is bypassed, and the algorithm repeats the search on the same tree (no pruning or refining) with a decreased threshold $\tau_{k+1}$ in the next epoch. The decreased threshold reflects a lowered estimate of $f(x^*)$ based on the absence of high-performing nodes in the current epoch.

The thresholds $\{\tau_k\}_{k \geq 1}$ are updated iteratively using a binary search to approach $f(x^*)$. For each epoch $k$, the algorithm maintains an interval $[a_k, b_k]$ which is believed to contain $f(x^*)$. The threshold $\tau_k$ is set to the mid-point of $[a_k, b_k]$. The initial interval $[a_1, b_1]$ is set to the known range $[a, b]$ of $f(x^*)$. At the end of epoch $k$, if no leaf node is identified as high-performing, we set $a_{k+1} = a_k - (b_k - a_k)/2$ and $b_{k+1} = b_k - (b_k - a_k)/2$, which leads to a decreased threshold in the next epoch. Otherwise, we set $a_{k+1} = \tau_k - c2^{-\alpha \rho_k/d+1}$ and $b_{k+1} = b_k$, where $\rho_k$ is the height of the tree before the pruning stage of epoch $k$, and $c \in (0, 1/2)$ is a hyperparameter (specified in Sec. 3.2.2).

We emphasize that while the proposed domain-shrinking approach conforms to an ever-growing tree, the algorithm can be implemented without storing the entire tree. The only information about the

tree that needs to be maintained is the set $\mathcal{D}_k$ of high-performing leaf nodes identified in the pruning stage of each epoch $k$. A pseudo-code of GP-ThreDS is provided in the supplementary material.

## 3.2 Identifying high-performing nodes

We now specify the local algorithm for identifying high-performing nodes in a given epoch $k$. Recall that $\mathcal{D}_k$ denotes the set of high-performing nodes identified in epoch $k$. Each node in $\mathcal{D}_k$ has grown $d$ levels and produced $2^d$ leaf nodes in the refining stage of epoch $k$. The objective of epoch $k + 1$ is to determine which of the $2^d |\mathcal{D}_k|$ newly grown leaves are high-performing nodes defined by $\tau_{k+1}$.

In epoch $k + 1$, the only portion of the tree that is of interest is the $|\mathcal{D}_k|$ subtrees, each of height $d$ with a root in $\mathcal{D}_k$. Our approach is to treat these subtrees separately, one at a time. We can thus focus on one subtree to describe the algorithm for identifying which of the $2^d$ leaves are high-performing. The terms root, node, and leaf all pertain to this subtree. We also omit the epoch index for simplicity.

### 3.2.1 A random-walk based search for high-performing nodes

A straightforward approach to identifying the high-performing nodes is to test each of the $2^d$ leaf nodes directly. This, however, results in a large number of samples at suboptimal points when the dimension $d$ is high. Our approach is inspired by the RWT (Random Walk on a Tree) algorithm recently proposed as a robust and adaptive algorithm for stochastic convex optimization [17, 18, 19].

Assume first there is exactly one high-performing node among the $2^d$ leaf nodes. The basic idea is to devise a biased random walk on the tree that initiates at the root and walks towards the high-performing node at the leaf level. As illustrated in Fig. 2 with $d = 2$, at a non-leaf node, the random walk can take one of three directions: towards the parent or one of the two children (the parent of the root is itself). The correct direction is to walk along the shortest path to the high-performing leaf node. With the subset relation encoded by the tree, this implies moving to the child containing a threshold-exceeding point or to the parent when neither child contains threshold-exceeding points. Hence, to guide the random walk, a local sequential test is carried out on the two children, one at a time, to determine, at a required confidence level, whether it is threshold-exceeding (see Sec. 3.2.2). The walk then moves to the first child identified as threshold-exceeding (if any) or to the parent otherwise. The confidence level of the local sequential test at each non-leaf node is only required to ensure the walk is correctly biased, i.e., the probability of walking in the correct direction is greater than $1/2$.

On reaching a leaf node, the algorithm enters the *verification* stage to determine whether this node is the high-performing leaf node. If the decision is no, it moves back to the parent of this leaf node, and the random walk resumes. If yes, the algorithm exits (under the assumption of a single high-performing leaf node). This decision can be made by carrying out the same local sequential test that guides the random walk at non-leaf nodes. The only difference is in the required confidence level. Given that a false positive at a leaf cannot be corrected due to exiting while a false negative only resumes the random walk (hence retractable in the future), the confidence level for a positive decision needs to be sufficiently high to ensure the overall regret performance, while a negative decision only needs to ensure the bias of the walk (as in the non-leaf nodes).

When the number of high-performing leaf nodes is unknown and arbitrary in $\{0, 1, 2, \ldots, 2^d\}$, multiple runs of the random walk are carried out to identify them one by one. In addition, a termination test on the root node is carried out before each run to determine whether there are still unidentified high-performing leaf nodes. See the supplementary for details along with a pseudo code.

We emphasize that the local test is carried out using only observations from the current visit to this node; observations from past visits are forgotten. This is to ensure the random-walk nature of the process for tight-analysis. A computational benefit is that the matrices being inverted to compute the posterior distribution are always small, improving the run-time efficiency of the algorithm.

### 3.2.2 The local sequential test

The last piece of the puzzle in GP-ThreDS is the local sequential test on a given node of a subtree. Given a node/region, $D \subseteq \mathcal{X}$, a threshold $\tau$, and a confidence parameter $\eta \in (0, 1)$, the local sequential test needs to determine, with a $1 - \eta$ confidence level, whether $D$ contains a point with function value exceeding $\tau$.

The test first builds a discretization of the region $D$, denoted by the set $D_g = \{x_i\}_{i=1}^{|D_g|}$. The set of points in $D_g$ are chosen to ensure that $\sup_{x \in D} \inf_{y \in D_g} \|x - y\| \leq \Delta$. A simple way to construct such a discretization is to use uniform grids parallel to the axes with a resolution small enough to satisfy the above constraint. The parameter $\Delta$ in epoch $k$ is set to $\Delta_k = (c/L)^{1/\alpha} 2^{-\rho_k/d}$ and is used to control the approximation of the function values in $D$. Recall that $L$ is the Hölder continuity constant while $c \in (0, 1/2)$ is a hyperparameter. The local test sequentially queries points in the set $D_g$ to locally estimate $f$.

To determine whether there exists a point $x \in D$ with $f(x) \geq \tau$, the test builds a pair of Upper and Lower Confidence Bounds using sequentially drawn samples and compares each of them to prescribed values. If the UCB goes below $\tau - L\Delta^\alpha$, indicating that the node is unlikely to contain a $\tau$-exceeding point, the test terminates and outputs a negative outcome. On the other hand, if LCB exceeds $\tau$, then this is a $\tau$-exceeding point with the required confidence level. The test terminates and outputs a positive outcome. If both the UCB and LCB are within their prescribed "uncertainty" range, the test draws one more sample and repeats the process. A cap is imposed on the total number of samples. Specifically, the test terminates and outputs a positive outcome when the total number of samples exceeds $\bar{S}(p, L\Delta^\alpha)$. A description of the test for $s \geq 1$ after being initialized with a point $x_1 \in D_g$ is given in Fig. 3. We would like to emphasize that the posterior mean and variance $\mu_{s-1}$ and $\sigma_{s-1}^2$ considered in the description below are constructed only from the samples collected during that particular visit to the current node.

---

- If $\max_{x \in D_g} \mu_{s-1}(x) - \beta_s(\eta)\sigma_{s-1}(x) \geq \tau$, terminate and output $+1$.
- If $\max_{x \in D_g} \mu_{s-1}(x) + \beta_s(\eta)\sigma_{s-1}(x) \leq \tau - L\Delta^\alpha$, terminate and output $-1$.
- Otherwise, query $x_s = \arg\max_{x \in D_g} \mu_{s-1}(x) + \beta_s(\frac{\delta_0}{4T})\sigma_{s-1}(x)$
- Observe $y_s = f(x_s) + \epsilon_s$ and use (2) and (3) to obtain $\mu_s$ and $\sigma_s$. Increment $s$ by 1.
- Repeat until $s < \bar{S}(\eta, L\Delta^\alpha)$.
  If $s = \bar{S}(\eta, L\Delta^\alpha)$, terminate and output $+1$.

---

Figure 3: The local sequential test for the decision problem of finding a $\tau$-exceeding point.

The parameter $\beta_s(\nu) := B + R\sqrt{2(\gamma_{s-1} + 1 + \log(1/\nu))}$ for $\nu \in (0, 1)$. $\gamma_t$ is the maximum information gain at time $t$, defined as $\gamma_t := \max_{A \subset \mathcal{X}: |A| = t} I(y_A; f_A)$. Here, $I(y_A; f_A)$ denotes the mutual information between $f_A = [f(x)]_{x \in A}$ and $y_A = f_A + \epsilon_A$. Bounds on $\gamma_t$ for several common kernels are known [20, 21] and are sublinear functions of $t$.

The cap $\bar{S}(\eta, L\Delta^\alpha)$ on the maximum number of samples is given by

$$\bar{S}(\eta, L\Delta^\alpha) = \min\left\{t \in \mathbb{N} : \frac{2(1 + 2\lambda)\beta_t(\eta)|D_g|^{\frac{1}{2}}}{(L\Delta^\alpha)\sqrt{t}} \leq 1\right\} + 1. \tag{4}$$

The cap on the total number of samples prevents the algorithm from wasting too many queries on suboptimal nodes. Without such a cap, the expected number of queries issued by the local test is inversely proportional to $|f(x_{D_g}^*) - \tau|$, where $x_{D_g}^* = \arg\max_{x \in D_g} f(x)$. Consequently, small values of $|f(x_{D_g}^*) - \tau|$ would lead to a large number of queries at highly suboptimal points when $f(x_{D_g}^*)$ is far from $f(x^*)$. The cap on the number of samples thus helps control the growth of regret at the cost of a potential increase in the approximation error. It also reduces the cost in computing the posterior distribution by limiting the number of queries at a node.

Note that when the sequential test reaches the maximum allowable samples and exits with an outcome of $+1$, it is possible that $f(x_{D_g}^*) < \tau$ (i.e., no $\tau$-exceeding points in $D_g$). Thus, $\tau$ may not be a lower bound for the updated belief of $f(x^*)$, as one would expect in the case of an output of $+1$ from the sequential test. However, using Lemma 2, we can obtain a high probability lower bound on $\tau - f(x_{D_g}^*)$. This additional error term is taken into account while updating the threshold as described in Sec. 3.1. The hyperparameter $c$ trades off this error with the size of the discretization.

The sequential test can be easily modified to offer asymmetric confidence levels for declaring positive and negative outcomes (as required for in the verification stage of the RWT search) by changing the confidence parameter in $\beta_s$. Details are given in the supplementary material.

We point out that the construction of the UCB is based on the UCB score employed in IGP-UCB [6]. It is straightforward to replace it with other types of UCB scores. The basic thresholded domain shrinking structure of the proposed algorithm is independent of the specific UCB scores, hence generally applicable as a method for improving the computational efficiency and regret performance of GP-UCB family of algorithms.

## 4 Performance Analysis

In this section, we analyze the regret and computational complexity of GP-ThreDS. Throughout the section, $D \subseteq \mathcal{X}$ denotes a node visited by GP-ThreDS, $D_g$ denotes its associated discretization, constructed as described in Sec. 3.2.2, and $x^*_{D_g} = \arg\max_{x \in D_g} f(x)$.

### 4.1 Regret Analysis

The following theorem establishes the regret order of GP-ThreDS.

**Theorem 1.** *Consider the GP-ThreDS algorithm as described in Sec. 3. Then, for any $\delta_0 \in (0,1)$, with probability at least $1 - \delta_0$, the regret incurred by the algorithm is given as*

$$R(T) = O(\sqrt{T\gamma_T}\log T(\log T + \sqrt{\log T \log(1/\delta_0)})).$$

We provide here a sketch of the proof. The regret incurred by GP-ThreDS is analysed by decomposing it into two terms: the regret in the first $k_0$ epochs referred to as $R_1$, and the regret after the completion of the first $k_0$ epochs referred to as $R_2$, where $k_0 = \max\{k : \rho_k \leq \frac{d}{2\alpha}\log T\}$. To bound $R_1$, we first bound the regret incurred at any node visited during the first $k_0$ epochs using the following decomposition of the instantaneous regret:

$$f(x^*) - f(x_t) = [f(x^*) - \tau_k + L\Delta_k^\alpha] + [\tau_k - f(x^*_{D_g}) - L\Delta_k^\alpha] + [f(x^*_{D_g}) - f(x_t)].$$

In the above decomposition, $k$ denotes the epoch index during which the node is visited. Each of these three terms are then bounded separately. The third term in the expression is bounded using a similar approach to the analysis of IGP-UCB [6] (notice that $x_t$ is the maximizer of the UCB score) that is to bound it by the cumulative standard deviation ($\sum_{s=1}^t \sigma_{s-1}(x_s)$).

**Lemma 1.** *For any set of sampling points $\{x_1, x_2, \ldots, x_t\}$ chosen from $D_g$ (under any choice of algorithm), the following relation holds: $\sum_{s=1}^t \sigma_{s-1}(x_s) \leq (1+2\lambda)\sqrt{|D_g|t}$, where $\sigma_s(x)$ is defined in (3).*

Since GP-ThreDS ensures a constant-sized discretization at all times (See Lemma 4), the above lemma implies that the sum of posterior standard deviations is $O(\sqrt{t})$ resulting in a tight bound corresponding to the third term (that is an $O(\sqrt{\gamma_t})$ tighter than the bound for IGP-UCB which optimizes the UCB score over the entire domain). The first two terms are bounded using the following lemma with an appropriate choice of $\Delta_f$.

**Lemma 2.** *If the local test is terminated by the termination condition at instant $\bar{S}(\delta_2, \Delta_f)$ as defined in (4), then with probability at least $1 - \delta_2$, we have $\tau - L\Delta^\alpha - \Delta_f \leq f(x^*_{D_g}) \leq \tau + \Delta_f$.*

The final bound on $R_1$ is obtained by a combination of the upper bound on regret on each node and the bound on the total number of nodes visited by GP-ThreDS, captured in the following lemma.

**Lemma 3.** *Consider the random walk based routine described in Section 3.2 with a local confidence parameter $p \in (0, 1/2)$. Then with probability at least $1 - \delta_1$, one iteration of RWT visits less than $\frac{\log(d/\delta_1)}{2(p-1/2)^2}$ nodes before termination.*

To bound $R_2$, we bound the difference in function values using the Hölder continuity of the function along with the upper bound on the diameter of the nodes after $k_0$ epochs. Adding the bounds on $R_1$ and $R_2$, we arrive at the theorem. The detailed proofs are provided in the supplementary material. We would like to point out that the regret analysis depends on the choice of the UCB score. While we have used the UCB score of IGP-UCB, this analysis is straightforward to extend to other UCB scores.

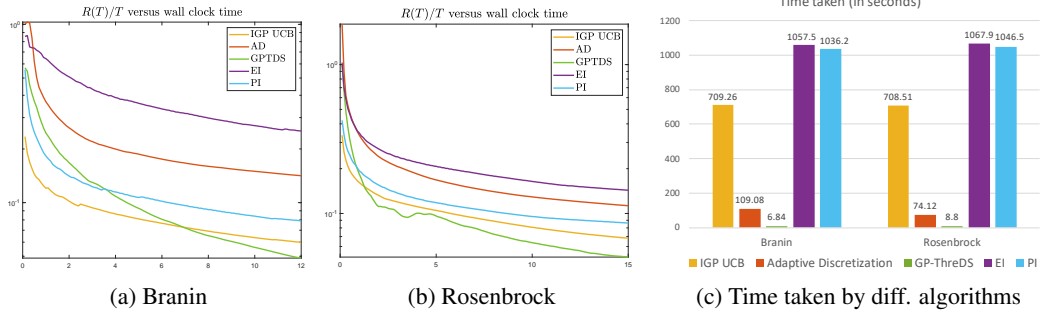

Figure 4: (a)-(b) Average cumulative regret against wall clock time for different algorithms on benchmark functions. (d) Computation time (in seconds) for 1000 samples for different algorithms.

**Remark 1.** *We note that our assumptions are consistent with those used in proving the lower bounds. In particular, the lower bounds are proven for the Matérn family of kernels including the SE kernel in [4]. [13, Proposition 1] proves the Hölder continuity of this family of kernels. Thus, our assumption on Hölder continuity is consistent with the lower bound. In addition, the proof of lower bound considers a class of functions whose RKHS norm is upper bounded by a known constant [4, Sec 1.1]. This upper bound translates to an upper bound on the absolute value of $f$, which is consistent with our assumption on having a finite range for $f$.*

### 4.2 Computational Complexity

The following theorem bounds the worst-case overall computational complexity of GP-ThreDS.

**Theorem 2.** *The worst-case overall computational complexity of GP-ThreDS is $O(T^4)$, where $T$ is the time horizon.*

The proof of theorem follows from the following lemma.

**Lemma 4.** *The number of points in the discretization, $|D_g|$, for any node $D$, is upper bounded by a constant, independent of time. i.e., $|D_g| = O(1)$, $\forall\, t \leq T$.*

From the lemma, we can conclude that the number of UCB score evaluations in GP-ThreDS is constant at all times $t$, hence matrix inversion becomes the dominant source of computational complexity. Since no more than $t$ samples are used to compute the posterior distribution at time $t$, the worst-case cost associated with matrix inversion step is $O(t^3)$ and consequently the worst-case computational complexity of GP-ThreDS is $O(T^4)$ leading to computational savings of $O(T^{2d-1})$ over GP-UCB family of algorithms. Lemma 4 is proven by showing that the rate of domain size shrinking matches the rate of granularity of the discretization across epochs. Thus, the size of discretization does not need to increase with time. Please refer to the supplementary for a detailed proof.

While the discretization does not grow with $t$, it is exponential in $d$. Since non-convex optimization is NP-Hard, such an exponential dependence on $d$ is inevitable for maintaining the optimal learning efficiency. In this work, we focus on reducing the computational complexity with respect to the time horizon $T$. The proposed domain shrinking technique can be used in conjunction with dimension reduction techniques (e.g., [22]) to achieve efficiency in both $T$ and $d$ (although at the price of invalidating the regret bounds).

## 5 Empirical Studies

In this section, we compare the performance of GP-ThreDS with several commonly used Bayesian optimization algorithms: IGP-UCB [6], Adaptive Discretization (AD) [23], Expected Improvement (EI) [24] and Probability of Improvement (PI) [25]. For the local test of GP-ThreDS we use the exact same UCB score as the one in IGP-UCB.

We compare these algorithms on two standard benchmark functions for Bayesian optimization: *Branin* and *Rosenbrock* (see [26, 27] as well as the supplementary material for their analytical expressions). We use the SE kernel with lengthscale of $l = 0.2$ on domain $[0,1]^2$. We use a Gaussian noise with

variance of 0.01. The parameters $\lambda$ in the GP model and $R$ in $\beta_t$ are also set to 0.01. The value of $\delta_0$ is set to $10^{-3}$. To limit the computational cost in the standard implementation of IGP-UCB, we consider a maximum of 6400 points in the grid.

Figures 4a, 4b, show the per-sample average regret, in log scale, measured at every 0.1 seconds, wall clock time. Specifically, within a given time (the X-axis of the figures), different algorithms process different number of samples determined by their computational complexity. The average per sample regret is then shown against the time taken. The plots are the average performance over 10 Monte Carlo runs. As expected from theoretical results, GP-ThreDS achieves the best performance especially as time grows. Figure 4c directly compares the computation time of all algorithms for processing 1000 samples, averaged over 10 Monte Carlo runs. GP-ThreDS enjoys a much smaller computation cost in terms of time taken (in seconds).

The details of algorithm parameters, benchmark functions, as well as additional experiments on hyper-parameter tuning of a convolutional neural network for image classification are in the supplementary.

# 6 Conclusion

A GP-based algorithm witha regret of $\tilde{O}(\sqrt{T\gamma_T})$ for black-box optimization under noisy bandit feedback was proposed. That is order optimal, up to poly-logarithmic factors, for the cases where a lower bound on regret is known. The proposed approach is rooted in the methodology of domain shrinking realized through a sequence of tree-based region pruning and refining to concentrate queries in high-performing regions of the function domain. It offers high learning efficiency, allows tight regret analysis, and achieves a computational saving of $O(T^{2d-1})$ over GP-UCB family of algorithms.

## Acknowledgments and Disclosure of Funding

The work of Sudeep Salgia and Qing Zhao was supported by the National Science Foundation under Grants CCF-1815559 and CCF-1934985. The work of Sattar Vakili was supported by MediaTek Research. We would also like to thank the anonymous reviewers for their constructive comments.

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
