# OpenReview forum: "A Domain-Shrinking based Bayesian Optimization Algorithm with Order-Optimal Regret Performance"
_NeurIPS.cc/2021/Conference — NeurIPS 2021 Poster_

### Official Review · Reviewer_GKzd · 2021-06-28

**Rating:** 7
**Confidence:** 4

**Summary:**

This paper proposes a GP based Bayesian optimization algorithm for regret minimization. This paper uses a domain-shrinking approach where the domain is explored as a tree. Using smoothness assumptions on the true function, it is possible to get upper and lower bounds on the function value in an entire region and this allows the algorithm to decide whether to subdivide a region to explore further or remove it entirely. A key idea and contribution of this work is that the number of leaves in the tree at any one time is a constant. Hence, the total computational complexity of this algorithm is reduced greatly compared to the standard complexity incurred by the famed GP-UCB algorithm. Additionally, the algorithm achieves order optimal performance, closing a gap of \sqrt(\gamma_t) observed in previous work. The paper closes with simple numerical experiments.

**Limitations And Societal Impact:**

Not discussed in this paper. Note that the submission guidelines state that a paper being theoretical is not sufficient grounds to claim that there are no societal impacts: https://neurips.cc/Conferences/2021/PaperInformation/PaperChecklist. A proper discussion should be added, though I don’t think anything extensive is needed.



**Main Review:**

Strengths: The paper is very well written. The algorithm, which complicated, is clearly stated and easily understood. The theoretical performance is strong as well and the paper closes a known gap. Furthermore, the computational gain is large.

Weaknesses: Error bars are omitted from the experiments in the main paper and moved to the appendix where they are somewhat large. Presumably this can be fixed by running more repetitions for a final draft. Axis and legend labels on the figures should also be larger to be more legible. Additionally, it is somewhat hard to interpret the experimental results. The results are stated in terms of wall clock time, but no significant discussion of the implementations is given, though this can have a large impact on wall clock performance. Additionally, the theoretical results of this paper state a gain in terms of the regret bound. However, as the results are given for wall-clock time, it is not possible to evaluate this empirically- though this seems like a significant contribution of this paper.

Other comments and questions:
-	Saying that the prior is fictitious is somewhat misleading. In terms of implementing the algorithm that is true, but all of the analysis assumes that the prior is real and so does this work.
-	For the statement of Thm 1, \delta_0 is presumably user provided? Please clarify.


**Time Spent Reviewing:**

3.5

---

> ### Author Response · Authors · 2021-08-10
> **Response to Reviewer GKzd**
>
> Thank you for your positive feedback on our paper. We are glad that you find the paper well written and clear, and the results strong. We will respond to your comments below.
>
> - *"Error bars are omitted from the experiments in the main paper and moved to the appendix where they are somewhat large. Presumably this can be fixed by running more repetitions for a final draft. Axis and legend labels on the figures should also be larger to be more legible."*
>
> Following your suggestion, we will increase the number of Monte Carlo runs in the experiments. Due to space constraints in the main text, we used a smaller size for the figures and removed the error bars for clarity. If the paper is accepted, we will have an extra page in the camera ready version which we will use to increase the size of the figures.
>
> - *"Additionally, it is somewhat hard to interpret the experimental results. The results are stated in terms of wall clock time, but no significant discussion of the implementations is given, though this can have a large impact on wall clock performance."*
>
> Thanks for your comment. The wall-clock running time of the algorithm is computed as the total time taken to find a query point minus the time taken to compute the function values (as the values of f are immediately observable). We will exapnd our discussion on this in Appendix D.3.
>
> Additional details of the devices used for the experiments are given in Appendix D.3.
>
> - *"Additionally, the theoretical results of this paper state a gain in terms of the regret bound. However, as the results are given for wall-clock time, it is not possible to evaluate this empirically- though this seems like a significant contribution of this paper."*
>
> We chose regret vs wall-clock time plots because that reflects both favorable aspects of GP-ThreDS: sample complexity (regret) and computational complexity which we think provides a fair comparison with the benchmarks. It is, however, not difficult to plot regret vs number of samples. We will add these figures to the camera ready version.
>
> - *"Saying that the prior is fictitious is somewhat misleading. In terms of implementing the algorithm that is true, but all of the analysis assumes that the prior is real and so does this work."*
>
> Our assumptions on the model are that $f$ is fixed and belongs to an RKHS, and noise is sub-Gaussian. The use of prior GP model is only for algorithm design. Our analysis also only assumes that $f$ is fixed and belongs to an RKHS, and noise is sub-Gaussian. In this sense, we refer to the prior GP model as fictitious because it is internal to the algorithm and is not considered neither in our problem formulation nor in the analysis. For the clarity of notation, we have used $F$ to distinguish the surrogate GP model from the true model $f$.  The randomness in the statement of our regret bound is only with respect to the observation noise, and not the prior GP model.
>
> Our regret bound holds for any $f$ in the RKHS. We agree that it is in contrast to an alternative (and typically simpler to analyze) formulation of the problem where $f$ is considered to be a sample from a GP. In the latter case, the regret bounds are given with respect to the randomness in $f$ (either in expectation or in high probability) in addition to the randomness in noise.
>
> Both of the formulations mentioned above have been considered in the literature. For example the analysis of GP-UCB like algorithms in [5,6,13] is under the exact same formulation as ours (fixed and unknown $f$ in RKHS and a fictitious GP model for algorithm design, the regret bounds hold for any fixed $f$ in the RKHS). The analysis of GP-UCB in [1] however considers both formulations. Their Theorems 2 and 3 consider the two cases of the random $f$ (as a sample from a GP) and the fixed $f$ (in an RKHS), respectively.
>
> We will highlight that the randomness in the statement of our regret bound is only with respect to the observation noise, and not the prior GP model, similar to [5,6,13].
>
> - *"For the statement of Thm 1, $\delta_0$ is presumably user provided? Please clarify."*
>
> Yes, $\delta_0$ is a user-specified value which determines the confidence level. We will make this clear in the revised paper.

---

> > ### Comment · Reviewer_GKzd · 2021-08-31
> > **Thank you**
> >
> > Thank you for your response and for answering my questions. On the whole, my opinion remains the same that this is a strong theoretical paper that will benefit from an extra page and some extra time for its experiment section. I'd definitely appreciate seeing a regret vs. # of samples plot in the final version.

---

### Official Review · Reviewer_HjBm · 2021-07-16

**Rating:** 7
**Confidence:** 4

**Summary:**

This paper introduces a GP optimization algorithm under the RKHS setting. The algorithm uses a tree-based domain shrinking approach where elimination, region splitting, and threshold updating are conducted in each epoch. This algorithm achieves a regret guarantee of $O(\sqrt{T\gamma_T})$ and has a low computational complexity of $O(T^4)$.

**Limitations And Societal Impact:**

Main comments:
- Related Work: More should be said about other domain-shrinking algorithms that work under general smoothness assumptions, such as SOO — could these potentially also give optimal bounds? Some ideas might even be related to those in algorithms like DIRECT (e.g., splitting the longest edge).
- More importantly, the idea of using a local GP in a small region was already used in [5], and so [5] seems potentially very closely related, but is only mentioned in passing. Please provide a clear comparison in the responses and also in the final paper, both in Section 1.3 and in any relevant technical sections.
- Related Work: The paper "High-Dimensional Experimental Design and Kernel Bandits” (ICML 2021) should also be mentioned, since it gets the same bound as SupKernelUCB but could be more practical (this hasn’t been verified yet)
- It would be better to mention how to construct a discretization in the main paper.
- Around Figure 3 I don’t see an explicit mention of whether $\mu_{s-1}$ and $\sigma_{s-1}$ are computed using all samples so far, or only the ones at the current node — this should be highlighted more clearly, as it seems to be fundamental.
- The proof of Lemma 3 is a bit hard to follow, especially regarding the definition of $\tau_d$, which could be reworded.
- If the authors are claiming optimality, they should make sure that any extra assumptions are consistent with the proofs of the lower bounds.  This should be commented on for Holder continuity and f in [a,b]. Both are consistent based on my understanding.
- The authors mention order-optimality in many parts, including the title. This ignores log factors, but that should be OK because it’s clarified early in the abstract. But another issue is that matching lower bounds are only known for a few kernels, not general kernels. Hence, the phrasing should be more modest to reflect this.
- Figure 4: Can error bars be added?
- Supplementary: Some notation, e.g., $\bar{S}(p)$, can be hard to locate. I suggest giving more reminders throughout, or even a table of notation.

Minor comments:
- Line 41: The phrasing “state-of-the-art regret bound of GP-UCB” is ambiguous, and makes it sounds (incorrectly) like that bound is the best known for any algorithm
- Line 49: Only use phrasing “needs to” for necessary conditions, not sufficient conditions
- Line 100-101: Phrasing “for the general case” isn’t so clear.  You could say “in other cases”, or more specifically mention the Matern kernel.
- In the main text, mention where the proofs can be found
- Line 328: Avoid the unusual phrasing “would have”
- The coefficients in some equations don't seem correct to me, but they don't affect the main result: appendix p9 -- the final step of $R^{(1)}$; appendix p10 -- the final step of $r_t$.
- Typos: appendix p9 -- $gamma_N$; appendix p16 -- additional minus sign in the first line of the second case.

**Main Review:**

This paper is well-written overall and easy to understand. The tree-based domain shrinking strategy presents its novelty over prior works. The nearly optimal regret guarantee is also a highlight. Most of the proofs seem correct and easy to follow. A comprehensive set of experiments shows that the proposed algorithm also performs well empirically. Some suggestions are given in the Limitations section below.

**Time Spent Reviewing:**

6

---

> ### Author Response · Authors · 2021-08-10
> **Response to Reviewer HjBm**
>
> Thank you for your positive feedback on our work. We are glad that you find the paper well written, the algorithm novel and the results strong. We also strongly appreciate your careful comments on the typos and grammar mistakes. We will correct them in the final version of the paper.
>
> Below, please find the response to your itemized comments.
>
> - *"Related Work: More should be said about other domain-shrinking algorithms that work under general smoothness assumptions, such as SOO — could these potentially also give optimal bounds? Some ideas might even be related to those in algorithms like DIRECT (e.g., splitting the longest edge). "*
>
> We have referenced SOO and HOO (that is similar to SOO except for the knowledge of Lipschitz continuity parameter) in the supplementary material (references [22] and [24] in the supplementary material), and acknowledged that a tree structure to represent successive partitions of the search domain is a classical approach and has seen its use in the bandit literature. Following your comment, we will expand on this discussion and mention that [13] and [22] which are referenced in the main paper can also be seen as the adaptation of SOO-type algorithms in our setting. In particular, while [22,24] among several other works consider (locally) Lipschitz continuous functions $f$, [13] and [22] consider the samples $f$ from a GP and the elements $f$ of an RKHS, respectively.
>
> We would like to point out that despite the apparent similarity of GP-ThreDS with SOO-type algorithms in using a tree structured search, there are substantial algorithmic differences. While GP-ThreDS uses a domain-shrinking methodology realized through a sequence of tree-based region prunings in conjunction with an outer loop of updating a threshold, SOO-type algorithms typically traverse the tree starting from the root in the search for a global maximum of a UCB index in each iteration. In this sense SOO-type algorithms do not perform a domain shrinking like GP-ThreDS.
>
> As it comes to the regret bounds, we compare our results with LP-GP-UCB [13] that is a SOO-type algorithm in the exactly same setting as ours ($f$ in an RKHS, noise being sub-Gaussian). [13] showed that the regret performance of LP-GP-UCB matches the lower bounds, in order, for some configuration of parameters $\nu$ and $d$ in the case of a Matérn kernel; in particular, when $\nu\le\frac{d(d+1)}{2}$. Under the general case, however, the regret bounds proven for LP-GP-UCB show a polynomial gap with the lower bounds. In addition, LP-GP-UCB seems impractical due to large constant factors, although a practical heuristic was also given in [13]. It is unclear to us whether another variation of SOO-type algorithms can achieve optimal order regret bounds in the RKHS setting.
>
> With respect to the DIRECT algorithm, to the best of our knowledge the DIRECT algorithm is specialized for the noise-free optimization of Lipschitz continuous functions. There are similarities with GP-ThreDS in terms of pruning the domain and focusing on high-performing regions. There are however also clear distinctions in the complexities that GP-ThreDS faces due to noisy observations and the RKHS structure which it handles using local tests and a random walk module.
>
> - *"More importantly, the idea of using a local GP in a small region was already used in [5], and so [5] seems potentially very closely related, but is only mentioned in passing. Please provide a clear comparison in the responses and also in the final paper, both in Section 1.3 and in any relevant technical sections."*
>
> Thank you for putting emphasis on [5]. The paper introduces $\pi$-GP-UCB, a variant of GP-UCB that partitions the domain into increasingly many hypercubes and fits an independent GP model to each hypercube. As a member of the GP-UCB family of algorithms, $\pi$-GP-UCB faces the same challenges stated in Section 1.1. First, the regret bound of $\pi$-GP-UCB, despite being better than GP-UCB, is not order optimal. Note that $\pi$-GP-UCB is specialized for the Matérn family of kernels. When compared on Matérn kernels, the regret bound of $\pi$-GP-UCB is uniformly (in all values of $\nu\ge0.5$ and $d\ge 1$) worse than that of GP-ThreDS. Second, the algorithm also faces the challenges of maximizing the UCB index over the entire domain making the algorithm computationally expensive.
>
> Following your comment, in the camera ready version of the paper, we will expand our discussion of [5].
>
> - *"Related Work: The paper "High-Dimensional Experimental Design and Kernel Bandits” (ICML 2021) should also be mentioned, since it gets the same bound as SupKernelUCB but could be more practical (this hasn’t been verified yet)"*
>
> Thank you for bringing this work to our attention. We were not aware of this paper which seems to be posted to ArXiv a few days before the NeurIPS submission deadline.
>
> This papers seems to also provide optimal order regret bounds for the same problem as ours. They are however using a different regression method compared to the standard Gaussian process or kernel ridge regression methods. Their proposed algorithm is based on an arm elimination technique on a finite domain which is rooted in the arm elimination algorithms for the $K$-armed bandit problems (e.g. see Auer and Ortner 2011, "UCB Revisited ..."). The algorithm also has similarities to SupKernelUCB in partitioning the past observations into independent batches.
>
> Although this is not proven in their paper, perhaps a discretization argument can extend their results to continuous domains. The size $K$ of discretization can be determined based on a priori known time horizon $T$ in a way that preserves the regret bounds. A fine discretization in the RKHS setting requires $K=O(T^{2d})$, implying an at least $O(T^{2d})$ computational complexity. Thus, this arm elimination based algorithm applied to a continuous domain will also incur an $O(T^{2d})$ computational cost, which is similar to GP-UCB based algorithms. Thus, although this work addresses the problem of order optimal regret bounds, their computational complexity remains prohibitively large in continuous domains.
>
> We will add a remark on this work. Thanks.
>
> - *"It would be better to mention how to construct a discretization in the main paper."*
>
> For domains that are in $\mathbb{R}^d$, simple uniform grids parallel to the euclidean axes can be considered.  The only requirement is a bound on the fill distance (the maximum distance of a node in the domain to the grid) which can be easily satisfied by a carefully selecting the resolution for the grid. We will specify this in the main paper for clarity.
>
> - *"Around Figure 3 I don’t see an explicit mention of whether $\mu_{s-1}$  and $\sigma_{s-1}$  are computed using all samples so far, or only the ones at the current node — this should be highlighted more clearly, as it seems to be fundamental."*
>
> It is computed only using the samples obtained during that particular visit to the node. Thank you for pointing this out. We will emphasize this in the figure as well as in main text.
>
> - *"The proof of Lemma 3 is a bit hard to follow, especially regarding the definition of $\tau_d$, which could be reworded."*
>
> Thank you for letting us know. We will revise the proof and ensure that the wording is easier to follow.
>
> - *"If the authors are claiming optimality, they should make sure that any extra assumptions are consistent with the proofs of the lower bounds. This should be commented on for Holder continuity and $f \in [a,b]$. Both are consistent based on my understanding."*
>
> Thank you for pointing this out. Our assumptions are consistent with those used in proving the lower bounds. Thus, our claim of order optimality is justified. In particular, the lower bounds are proven for the  Matérn family of the kernels including the squared exponential kernel [4]. [13, Proposition 1] proves the Hölder continuity of this family of kernels. Thus, our assumption on Hölder continuity is consistent with the lower bound. In addition, the proof of lower bound considers a class of functions whose RKHS norm is upper bounded by a known constant [4, Sec. 1.1]. This upper bound translates to an upper bound on the absolute value of $f$ which is consistent with our assumption on having a finite range for $f$.
>
> We will add a remark on this in the final version of the paper.
>
> - "The authors mention order-optimality in many parts, including the title. This ignores log factors, but that should be OK because it’s clarified early in the abstract. But another issue is that matching lower bounds are only known for a few kernels, not general kernels. Hence, the phrasing should be more modest to reflect this."*
>
> Thanks for pointing out this presentation issue. We will make sure to add phrases "up to logarithmic factors" and "for the cases where the lower bound is known (i.e., with Matérn and SE kernels)" wherever appropriate to avoid misrepresentation of the results.
>
> - *"Figure 4: Can error bars be added?"*
>
> With a restriction the size of the figures in the main text due to space constraints, we had removed the error bars for clarity. We have, however, included the error bars in the figures in the supplementary material. If the paper is accepted, we will have an extra page in the camera ready version, which we can use to increase the size of the figures.
>
> - *"Supplementary: Some notation, e.g., $\bar{S}(p)$, can be hard to locate. I suggest giving more reminders throughout, or even a table of notation"*
>
> Thank you for pointing this out. We will make sure that the notation is clarified to ensure that proofs are easier to follow.

---

### Official Review · Reviewer_UWNk · 2021-07-16

**Rating:** 7
**Confidence:** 2

**Summary:**

The paper proposes a Guassian process-based Bayesian optimization algorithm for solving the black-box optimization problems with order-optimal regret guarantee. The main contribution of the paper is in domain-shrinking strategy implemented through tree-based region pruning and focusing on high-performing regions to offer high learning and computational efficiency.

The paper presents solid theoretical proofs of performance analysis, as well as empirical results on benchmark problems.


**Limitations And Societal Impact:**

Some limitations are already presented in Section 4. However, more discussion on when the method fails or performs worse than other algorithms shown in Figure 4 would improve the understanding of the method.


**Main Review:**

The paper is well-written, clear, and easy to follow. It solves an interesting problem with many potential applications. It is both theoretically sound and includes good empirical studies.

A few comments and questions for potential improvement/clarification of the method:

Section 1.2: an illustration of the general idea (domain shrinking concept) would help in understanding the method. Potentially simply refer to Figure 1.

Figure 1: Extend the caption to explain what is presented in the figure and what each symbol stands for.

Could the algorithm be parallelized to improve the time efficiency?

Only 2 benchmark functions are shown in the results of the paper. More test problems would be appreciated.


**Time Spent Reviewing:**

5

---

> ### Author Response · Authors · 2021-08-10
> **Response to Reviewer UWNk**
>
> Thank you for the the positive feedback on our work. We are glad that you find the paper well written and the problem interesting. Below, please find the responses to your itemized comments.
>
> - *"Section 1.2: an illustration of the general idea (domain shrinking concept) would help in understanding the method. Potentially simply refer to Figure~1"*:
>
> Thanks for your suggestion. We agree that our high level presentation of the algorithm in Sec 3.1 can benefit from reference to the illustrations in Figures 1 and 2. We will add more details.
>
> - *"Figure~1: Extend the caption to explain what is presented in the figure and what each symbol stands for"*:
>
> Thank you for the suggestion. We will revise the paper accordingly.
>
> - *"Could the algorithm be parallelized to improve the time efficiency?"*
>
> Yes, the computation for refining the high-performing nodes (see Sec.~3.1) can be implemented in parallel across the different nodes, which will contribute to further improved runtime efficiency. In the runtime simulation results currently included in the paper, we have not considered parallel implementations.
> - *"Only 2 benchmark functions are shown in the results of the paper. More test problems would be appreciated"*:
>
> Thank you for the suggestion. We aim to include more benchmark functions in the revised paper.

---

### Decision · Program_Chairs · 2021-09-27

**Decision:**

Accept (Poster)

**Comment:**

The reviewers are all in agreement that this work is suitable for publication at NeurIPS, particularly due to being one very few works to attain near-optimal cumulative regret in kernelized bandits.  The reviewers also judged that their concerns were adequately addressed by the authors.  Although the comments/suggestions given are largely minor, I ask the authors to carefully consider them in the camera-ready version.  I particularly highlight the following regarding the related work:
- "[5] seems potentially very closely related, but is only mentioned in passing. Please provide a clear comparison in the responses and also in the final paper, both in Section 1.3 and in any relevant technical sections."
- "[the recent paper] "High-Dimensional Experimental Design and Kernel Bandits” (ICML 2021) should also be mentioned" (it is appreciated that this paper was only made public very close to the NeurIPS deadline)